# Increased EGFRvIII Epitope Accessibility after Tyrosine Kinase Inhibitor Treatment of Glioblastoma Cells Creates More Opportunities for Immunotherapy

**DOI:** 10.3390/ijms24054350

**Published:** 2023-02-22

**Authors:** Cezary Tręda, Aneta Włodarczyk, Marcin Pacholczyk, Adrianna Rutkowska, Ewelina Stoczyńska-Fidelus, Amelia Kierasińska, Piotr Rieske

**Affiliations:** 1Department of Tumor Biology, Medical University of Lodz, Zeligowskiego 7/9, 90-752 Lodz, Poland; 2Department of Research and Development, Celther Polska Ltd., Inwestycyjna 7, 95-050 Konstantynow Lodzki, Poland; 3Department of Systems Biology and Engineering, Silesian University of Technology, Akademicka 16, 44-100 Gliwice, Poland

**Keywords:** EGFRvIII, dimerization, glioblastoma, CAR-T, immunotherapy

## Abstract

The number of glioblastoma (GB) cases is increasing every year, and the currently available therapies remain ineffective. A prospective antigen for GB therapy is EGFRvIII, an EGFR deletion mutant containing a unique epitope that is recognized by the L8A4 antibody used in CAR-T (chimeric antigen receptor T cell) therapy. In this study, we observed that the concomitant use of L8A4 with particular tyrosine kinase inhibitors (TKIs) does not impede the interaction between L8A4 and EGFRvIII; moreover, in this case, the stabilization of formed dimers results in increased epitope display. Unlike in wild-type EGFR, a free cysteine at position 16 (C16) is exposed in the extracellular structure of EGFRvIII monomers, leading to covalent dimer formation in the region of L8A4–EGFRvIII mutual interaction. Following in silico analysis of cysteines possibly involved in covalent homodimerization, we prepared constructs containing cysteine–serine substitutions of EGFRvIII in adjacent regions. We found that the extracellular part of EGFRvIII possesses plasticity in the formation of disulfide bridges within EGFRvIII monomers and dimers due to the engagement of cysteines other than C16. Our results suggest that the EGFRvIII-specific L8A4 antibody recognizes both EGFRvIII monomers and covalent dimers, regardless of the cysteine bridging structure. To summarize, immunotherapy based on the L8A4 antibody, including CAR-T combined with TKIs, can potentially increase the chances of success in anti-GB therapy.

## 1. Introduction

Glioblastoma (GB) is the most common malignant brain tumor in adults and one of the most lethal of all cancers [1]. Current treatment options include surgery, radiotherapy, and chemotherapy, which result in a median survival of only 12–15 months [2]. Hence, understanding the molecular principles and signaling pathways involved in GB is critical for the development of more effective and targeted therapies for these patients.

GB is characterized by extensive intratumoral heterogeneity, as observed in single-cell RNA-sequencing studies [3]. A frequently observed phenomenon in GB is amplification of the EGFR gene, which occurs in 40–50% of all indicated cases and is associated with an excessive amount of EGFR protein in the membrane [4,5,6,7]. Of the cases of GB with EGFR amplification, about half are positive for a truncated version of EGFR (EGFRvIII for short) [8]. In addition to GBs, EGFRvIII is also found in cancers of the head and neck, breast, prostate, and intestines, as well as in non-small cell lung cancer; however, in these cases, this protein exhibits decreased prevalence and is, moreover, produced at lower levels [9,10]. This truncated form of receptor arises from the deletion of exons 2–7 from the wild-type gene, which encodes a part of the extracellular domain responsible for ligand binding. This change leads to the creation of a constitutively active protein characterized by spontaneous dimerization of monomers and evidently increased stability, with the mutated protein exhibiting an extended half-life compared with the wild-type [11]. These features manifest as an extraordinary signaling pattern [6], resulting in increased cell viability and the avoidance of apoptosis [9,12].

Despite intensified research and broad knowledge about the dimerization and ligand binding characteristics of wild-type EGFR, our understanding of the mechanism of truncated receptor activation, complex formation, and its dynamics still remains incomplete [13]. Dimerization constitutes one of the key factors widespread in the processes occurring in cells, playing a crucial role in the activation and regulation of a multitude of pathways [13,14,15]. Initially, Fan et al. indicated specific oncogenic signaling relationships between EGFR and EGFRvIII in glioblastoma [13], but further reports also indicated a homodimerization ability with cross-activation between EGFRvIII monomers [13,14,16]. This unusual characteristic of EGFRvIII is linked to a free, unbridged cysteine (Cys16), which was indicated for the first time by Ymer et al. [17]. It was shown that Cys16 is responsible for covalent dimerization between two monomers [14,15,17]. A point mutation leading to the substitution of Cys16 with serine causes a strong decrease in homodimer stability with little influence on EGFR kinase phosphorylation (although this observation is still considered controversial) [15,16,17]. Another issue is the role of EGFR kinase inhibitors in the receptor dimerization process. In the case of EGFRwt, it was found that TKIs elevate dimer formation with a simultaneous decrease in the levels of receptor phosphorylation [18,19]. Hence, it is crucial to verify whether this phenomenon also occurs in EGFRvIII [20].

The truncated extracellular part of EGFR is challenging to target in personalized therapy, but attempting to develop anti-EGFRvIII antibody-based therapy, such as CAR-T (chimeric antigen receptor T cells), seems to be a promising approach [12,21]. As a result of exon loss at position 6, between amino acid residues 5 and 274 of wild-type EGFR, a new glycine residue arises, which results in the creation of a unique epitope that can potentially be recognized by specific antibodies [22]. The efficacy of CAR-T therapy has been especially demonstrated in the treatment of B-cell acute lymphoblastic leukemia (B-ALL) and lymphomas [23,24]. However, the progress achieved in this area does not apply to CAR-T cell therapy for treating solid tumors, mainly due to difficulties experienced by CAR-T cells in trafficking and infiltrating into tumor tissue, the immunosuppressive state of the microenvironment, immune-related adverse events, the genetic instability of cancer cells, cell toxicity, and antigen specificity (summarized by Ma et al. 2019 [25]). EGFRvIII thus seems to be a perfect tumor-specific target. Hence, several mAbs directed against this receptor have been developed for diagnostic and further therapeutic purposes, such as DH8.3, 806, and L8A4, with the latter appearing to be the most promising [26,27]. Despite safety concerns and delivery difficulties for solid tumor treatment, CAR-T has already exhibited potential for targeting EGFRvIII in in vivo GB studies [21].

Unfortunately, EGFRvIII-positive GB cells represent a minority of tumor cells within EGFRvIII-positive glioblastomas. Neoplastic changes, such as colorectal or breast cancer, apparently contain a marginal percentage of EGFRvIII-positive cells with very low expression of this oncogene [28]. Moreover, we did not find data to support EGFRvIII as a marker of GB stem cells [29,30,31]. In this scenario, the final outcome after the elimination of the minority of GB cells is uncertain, and methods for increasing EGFRvIII density, as well as the percentage of EGFRvIII-positive cells or highly positive cells within existing cancer cells, become important. We hypothesize that for glioblastoma resistant to EGFR TKI treatment, the use of immunotherapies should result in a better response due to the described mechanism [32,33].

Taken together, we analyzed the dimer formation mechanism and found that Cys16 is not the only cysteine involved in this process. Based on in silico data, we conclude that a series of consecutive cysteines may increase the possibility of forming different homodimer 3D structures, resulting in the so-called plasticity of the extracellular part of EGFRvIII. Various cysteine–serine substitutions at positions Cys20, Cys35, Cys38, and Cys42 decreased the capacity for covalent dimer formation. We further analyzed whether the levels of EGFRvIII phosphorylation converge with the kinase activity only or if perhaps this is also regulated by dimer stability. Furthermore, we tested whether the antibody L8A4 is similarly effective against both EGFRvIII monomers and dimers in cases where the different disulfide bridge arrangement may hamper interaction between the antibody and epitope. Such intense analysis of epitope recognition with the EGFRvIII-specific antibody L8A4 together with TKI treatment raises the possibility for further improvement of the effectiveness of antibodies or antibody-based therapies. The results of this study indicate that the analyzed antibody can be successfully used not only in the diagnosis of EGFRvIII-positive GB patients but also as a basis for prospective and effective anticancer CAR-T therapy.

## 2. Results

### 2.1. Characterization of Research Model According to the Different EGFRvIII Expression Profiles

The uniqueness of the EGFRvIII extracellular domain led us to search for an effective EGFRvIII-specific antibody. Various commercially available antibodies did not exhibit the desired specificity in this context. However, to our surprise, the L8A4 antibody [26,27] is able to specifically recognize the EGFRvIII epitope and not EGFRwt. Initial immunocytochemical staining (ICC) of the lines DK-MG^parental^, DK-MG^low^, MDA-MB-468, and two primary GB1 and GB2 cell lines (Figure 1A) revealed the high specificity of L8A4–EGFRvIII binding. EGFRvIII-negative/EGFRwt-positive cell line MDA-MB-468 was not recognized by the L8A4 antibody (Figure 1A). For more detailed analysis, DK-MG^parental^ and its subpopulations DK-MG^extralow^, DK-MG^low^, and DK-MG^extrahigh^ (lines obtained from single-cell colonies as previously described [16,34,35]) were analyzed for EGFRvIII and EGFRwt expression using real-time PCR and Western blotting (Figure 1B,C). We noticed a pronounced difference in EGFRvIII expression between extralow/low and parental/extrahigh, with almost no difference in EGFRwt (Figure 1B,C). Furthermore, the specificity of L8A4 was confirmed in primary GB1 and GB2 cell lines that had low EGFRwt and high EGFRvIII expression (Figure 1D,E). The effectiveness of the L8A4 antibody was also analyzed in cell lines with an EGFRvIII-positive homogeneous population: DK-MG^extrahigh^ (Figure 1F,G) and AD293^vIII^ (Figure 1H,I), and compared with the A10 antibody, which recognizes both wild-type EGFR and EGFRvIII. Finally, a similar antibody comparison was performed on the MDA-MB-468 cell line that lacks EGFRvIII (Figure 2A,B).

### 2.2. Influence of Particular TKIs

Next, we analyzed the effect of tyrosine kinase inhibitors (TKIs), lapatinib, erlotinib, and afatinib, on DK-MG^extrahigh^ (Figure 1F,G), DK-MG^parental^ (Figure 2C,D), and AD293 exogenously expressing EGFRvIII (further referred to as AD293^vIII^; Figure 1H,I). The 1 h treatment produced a significantly higher fluorescence signal from EGFR in erlotinib- and afatinib-treated cells compared with the lapatinib-treated or control (DMSO-treated) groups (*p* ≤ 0.05). Moreover, the analysis of cell fractions indicated that the application of erlotinib increased the amount of EGFRvIII, especially in the membrane and, to a lesser extent, in the nuclear and cytoskeletal compartments (Appendix A). A similar statistically significant effect of increased fluorescence related to TKI treatment was visible for NCI-H1975 cells expressing EGFR with L858R and T790M mutations (Figure 2E,F), as well as AD293 exogenously overexpressing EGFRwt (Figure 2G,H). Interestingly, despite EGF having an insignificant effect on both EGFRwt and EGFR kinase-mutant H1975-bearing cell lines, EGFR-related fluorescence drastically increased in MDA-MB-468 cells (Figure 2A,B). However, upon EGF treatment, MDA-MB-468 cells clearly became visibly smaller, which, combined with their unchanged levels of EGFR protein, produced a false-positive peak in immunocytochemistry staining that we ignored. To conclude, the TKI-related fluorescence increase in stained EGFR was observed to the same extent in EGFRvIII (DK-MG^parental^, DK-MG^extrahigh^, and AD293^vIII^) and EGFRwt/EGFR^L858R/T790M^ (MDA-MB-468, AD293^wt^, and NCI-H1975; Figure 2A,B,E–H).

### 2.3. Dimer Formation and EGFR Variant

In order to verify whether dimer formation is dependent on the activation of EGFRvIII, we used sodium orthovanadate (SOV), a compound proven to induce robust EGFRvIII phosphorylation [16]. The ability of EGFRwt and EGFRvIII to dimerize in AD293 cells was examined by Western blot in semi-native conditions (Figure 3A), which revealed that EGFRvIII, unlike EGFRwt, forms and accumulates as a stable homodimer, although the monomeric form is still visible. Accumulation of EGFRvIII in the form of a dimer was further confirmed in DK-MG^extrahigh^ (Figure 3B,C), additionally confirming that, unlike erlotinib and afatinib, lapatinib does not support dimer accumulation (Figure 3B,C). A side-by-side comparison of DK-MG and AD293 cell lines allowed us to confirm that the effect of erlotinib is similar in these cell lines. Importantly, we noticed that the EGFRvIII-specific antibody L8A4 recognizes dimers with the same specificity as the anti-total EGFR A10 antibody (Figure 3D,E). To exclude the influence of cell culture type on our analyses, we examined the cell culture in three different conditions: monolayer, suspension, and spheres. We did not observe any significant difference between the A10 and L8A4 antibodies in these cases (Appendix A), which was statistically confirmed (Appendix A). This led us to speculate that a part of EGFRvIII that participates in covalent dimerization is still available for the L8A4 antibody (Figure 3D,E). The addition of L8A4 to the cell culture has a visible influence on dimer formation ability (Appendix A). Furthermore, Ymer et al. [17] indicated that cysteine 16 is responsible for the stable (covalent) dimer formation of EGFRvIII. Thus, we analyzed the AD293 cell line expressing either EGFRvIII or EGFRvIII^C16S^ (C16S—cysteine to serine substitution) and noticed its strong effect in suppressing covalent dimer formation (Figure 3F). C16S caused a significant decrease in the ability to dimerize as well as slightly lower accumulation of the receptor compared with EGFRvIII, even upon either erlotinib or afatinib treatment (Figure 3F–H).

Factors positively or negatively influencing dimer formation are summarized in Figure 3I. Our interest is focused on the C16S substitution in EGFRvIII (unpaired). However, the question is whether one should only consider this particular cysteine in terms of the impairment of covalent dimerization. Understanding the mechanics of the extracellular truncated domain could broaden the knowledge of EGFRvIII and be fundamental to future therapy design.

### 2.4. In Silico Analysis of Dimer Formation

The thiol groups of cysteines can be divided into four groups depending on their reactivity: forming stable structural disulfide bonds, coordinating metals, remaining in an oxidized state, and being susceptible to reversible oxidation [36]. We carried out an analysis to determine the possibility of alternative disulfide bridge configurations within the fragment of domain II found in EGFRvIII. The premise for the analysis was the observation of five cysteines (Cys16, Cys20, Cys35, Cys38, and Cys42) in close proximity and the supposition that in EGFRvIII, a fragment of domain II (Figure 4A) may fold differently to the corresponding region in EGFRwt. The concept that differently folded forms of EGFRvIII may occur simultaneously in the cell (i.e., with free cysteines Cys16, Cys20, Cys35, Cys38, and Cys42) seems unlikely; however, rearrangement of bridges is an enzymatic process carried out by disulfide isomerase [37] and cannot be fully excluded.

We analyzed different possibilities for disulfide bridging of the domain II cysteines at both sequence and structure levels. Structural-level modeling was carried out in the MODELLER program [38]. Homology models were created based on the EGFRwt dimer structure (PDB ID: 3NJP). Disulfide bridges were explicitly and separately defined for each model (Appendix A). Our models show the possibility of different arrangements of domain II cysteine bridges (Figure 4B).

We were unable to obtain reliable dimer models from rigid-body protein–protein docking (Haddock, ClusPro). The EGFR extracellular domain, spanning over 600 amino acids, has two functionally important conformations (or states): the inactive and autoinhibited or tethered (T) monomer and the ligand-bound, active, or untethered (UT) form, which promotes EGFR dimerization. Despite the large deletion (exons 2–7) in the EGFRvIII mutant, the hinge region responsible for the conformational transition between T and UT states remains intact. In our opinion, the homology model based on the EGFR homodimer (PDB ID: 3NJP) does exhibit a conformation suitable for covalent homodimerization through Cys16 (or any other domain II cysteine—Cys20, Cys35, Cys38, or Cys42). Covalent EGFRvIII homodimerization is most likely a dynamic process that requires some conformational adaptation of the ectodomain in the hinge region.

Using a relatively simple cross-linking procedure (connecting and fusing in Schrodinger Maestro) followed by global model energy minimization in the OPLS forcefield, we were able to create homodimer models covalently linked by disulfide bridges between Cys16 (Figure 4C), Cys20 (Figure 4D), and Cys35 (Figure 4E). Such models, however, can only be regarded as an illustration of a certain idea, as there are serious steric clashes between subunits in some cases.

On the sequence level, we performed meta-analysis using six predictive algorithms. The programs operate on the following principle: based on the input sequence of amino acids, they predict the state of cysteines (bonded or free-non-bonded). The prediction is supported by models that were created by applying machine learning methods to known crystal structures from the PDB database. The algorithms differ in the details of the algorithmic approach used to learn and make decisions. DiANNA [39] and SCRATCH [40], as well as DBCP [41] and CYSCON [42], predict binding between cysteines, while DICON [42] and DISULFIND [43] only predict the cysteine state. DBCP additionally determines the likelihood of cysteine bonding with metals (Appendix A).

The consensus scoring procedure (Appendix A) indicates Cys16 as the most probable free cysteine, but others were not excluded. Encouraged by the in silico study, which linked particular cysteine locations and their engagement in the creation of disulfide bridges, we conducted site-directed mutagenesis of EGFRvIII cDNA, wherein we substituted five consecutive cysteines in the EGFRvIII structure with serines (Cys16, Cys20, Cys35, Cys38, and Cys42), which is visualized on a schematic picture of the introduced mutations (Figure 4A). Expression of these EGFRvIII cysteine mutants was characterized by real-time PCR (Figure 4F).

### 2.5. Validation of In Silico Model

Functional analysis by semi-native WB revealed that the absence of any cysteines (substituted by serine) clearly results in decreased dimer stability, which was demonstrated by blotting (Figure 5A). This means that all examined cysteines are independently responsible for covalent homodimerization. The above observation was not dependent on treatment conditions (DMSO, SOV, and erlotinib). Dimers were not formed in EGFRvIII phosphorylated (SOV-treated) samples, and, as expected, erlotinib inhibited SOV activity, although the formation of EGFRvIII dimers was still supported (Figure 5A).

Such a wide spectrum of cysteines is critical for covalent dimerization and, hence, substantial plasticity in bridge formation may constitute, once again, an obstacle to the activity and effectiveness of the L8A4 antibody. We compared the binding of the A10 and L8A4 antibodies for the whole spectrum of cell lines not only by ICC on AD293 bearing either EGFRvIII^C16S^ or EGFRvIII^C35S^ (Figure 5B,C), but also by WB (Figure 5D,E). The same conclusion was supported by both types of analysis: none of the cysteine substitutions influence L8A4 affinity to EGFRvIII. Densitometric analysis of blots clearly indicates that the dimer/monomer ratio remains as assessed by both antibodies (Figure 5F).

In summary, all 5 cysteine substitutions (as shown in the schematic in Figure 5G) hinder covalent dimer formation, but the EGFRvIII epitope remains available for detection by the L8A4 antibody. These data suggest that the unique epitope recognized by L8A4 is not involved in the conformational changes related to the dimerization process.

## 3. Discussion

Although EGFRvIII is found in about 27% of glioblastomas [28], due to difficulties in the long-term in vitro maintenance of cells bearing EGFRvIII [44], we focused on immortalized cell lines. A comparison of various cell lines showed we could distinguish between cells with EGFRwt and those with EGFRvIII. We noticed that different TKIs did not have the same effect on covalent dimer formation (erlotinib/afatinib vs. lapatinib), which opens up opportunities for using TKIs not only directly in treatment but also raises the question of whether TKIs can indirectly support CAR-T-based treatment or impose a problem for such therapy by decreasing the availability of the epitope.

Our doubt was due to the recently described mechanism of EGFR accumulation upon blocking the active conformation of the EGFR tyrosine kinase domain (possibly by erlotinib, gefitinib, or afatinib), which was not the case for lapatinib, which selectively binds the inactive configuration (our data and [45,46]). It was already observed in the case of wild-type EGFR that the dissociation of EGF is slower for EGFR dimers compared with monomeric EGFR, and TKI (erlotinib, gefitinib) treatment of dimers reduces the overall dissociation rate, followed by perturbations of internalization and slower degradation [46]. Our data indicate that TKIs not only caused an increase in EGFRvIII dimer stability but also increased the fluorescence signal corresponding to EGFR in cells, which suggests decreased internalization, lower degradation, or increased recycling of the receptor. This supports our hypothesis that some TKIs may increase (EGFRvIII) antigen exposure on the cell surface, improving the outcome of CAR-T-based therapy. On the other hand, we found that lapatinib treatment not only resulted in lower dimer formation ability but also contributed to a reduction in overall EGFRvIII-related fluorescence signal in ICC (in the case of DK-MG^extrahigh^) or had no influence (DK-MG^parental^ and AD293vIII). This could be because lapatinib favors the formation of monomeric EGFR, thus leading to a reduction in the overall number of dimers [46]. A more probable explanation is that lapatinib, by binding to EGFRvIII in an inactive configuration, blocks the modification of the extracellular part, impairing the availability of cysteines responsible for covalent dimerization. A decrease in the number of dimers is followed by a shift in the balance between degradation and recycling toward receptor degradation. It must be added that we assessed this phenomenon in a limited number of different cell lines, so it is still possible that recycling may be influenced by the innate mechanisms of each cell type [16,47,48]. However, this phenomenon is not unique to EGFRvIII. The addition of TKIs to the H1975 cell line also induces an enhancement of EGFR fluorescence. In these cells, mutations of EGFR other than EGFRvIII are expressed, i.e., L858R and T790M, which lead to improved patient responses when treated with erlotinib [49]. However, an increase in EGFR fluorescence in H1975 cells analyzed by ICC is mostly due to intracellular accumulation of the inactive form of EGFR [18]. Although de Wit et al. [18] indicated a lack of accumulation of EGFRwt unless mutated, in our studies this phenomenon was found to be dependent on the research model (accumulation after TKI treatment was not detected in MDA-MB-468 with endogenous EGFRwt expression (Figure 2A,B). Additionally, a significant increase in protein was observed in NCI-H1975 (simultaneously expressed EGFRwt and mutations L858R/T790M (Figure 2E,F)) and AD293wt (exogeneous expression of EGFRwt (Figure 2G,H)). In our ICC results, in contrast to those of de Wit, a visible increase in fluorescence of EGFRvIII is observed after TKI treatment. This can be explained by the differences in mechanism since EGFRvIII is blocked in dimer form, and this is known to be a factor that slows down internalization and receptor degradation kinetics, which in turn are responsible for protein accumulation [50]. On the other hand, it is more probable that de Wit et al., in their study, used well-known EGFRwt-bearing HeLa cells, which could drastically change the outcome due to the possibility of heterodimerization and is described elsewhere [13]. This is why our results are not contradictory and create a space for the further application of erlotinib or afatinib in EGFRvIII-focused CAR-T therapy.

In an attempt to understand the mechanism behind EGFRvIII homodimerization, we began with a reanalysis of the cysteine 16 (Cys16) mutant proposed by Ymer et al. [17]. Since we confirmed their findings, we wondered whether only Cys16 is important for covalent dimerization or whether the involvement of other cysteines should be considered. Homodimer models with subunits covalently linked by disulfide bridges between Cys16, Cys20, and Cys35 only illustrate this idea conceptually, as in some cases, there would be either serious steric clashes between subunits or a domain IV arrangement significantly different from that of the EGFRwt homodimer known from the crystal structure (PDB ID: 3NJP). However, covalent EGFRvIII homodimerization is most likely a dynamic process that requires some conformational adaptation of the ectodomain in the hinge region. These promising modeling results convinced us to prepare constructs with five cysteine–serine substitutions in EGFRvIII (Cys16, Cys20, Cys35, Cys38, and Cys42) to cover all possibilities and questions. We showed that some of these substitutions (C16S, C35S, C38S, and C42S) do indeed have a negative influence on covalent homodimer formation, and all of the substitutions showed a lack of influence on receptor tyrosine kinase activity.

Since the abovementioned findings demonstrate the strong influence of the EGFRvIII extra domain structure, we analyzed the effectiveness of the EGFRvIII-targeting antibody L8A4, which specifically recognizes the mutant and not wild-type EGFR. We also showed that L8A4 recognizes both the EGFRvIII monomer and homodimer and that conformational changes around cysteines do not influence epitope recognition. The only cysteine that is under question is Cys20 since it did not behave in the same manner as Cys16 or others. We suspect that Cys20′s underperformance could be due to its unfavorable position, the result of which is that another cysteine is more likely to be involved in disulfide bond formation at the expense of Cys20. This is visible in the case of Cys38 and Cys42, which are much more abundantly expressed. In this regard, we conclude that extracellular domain modifications by TKIs raised the possibility of further increasing antibody availability regarding the use of CAR-T as a therapy method. Other researchers have also reported the high specificity of this antibody to the discussed oncogene. Yang et al. demonstrated that L84A antibody cross-reaction with EGFRwt is negligible, whereas binding to EGFRvIII is highly specific [26].

Implementation of CAR-T therapy for the treatment of hematological malignancies constitutes a significant breakthrough in cancer immunotherapy in recent years. However, most antigens targeted by CARs are not tumor “specific” but tumor “associated”. This means that tumors may have higher expression of these antigens but that they are also present in normal cells [51]. EGFRvIII is frequently analyzed as a target for CAR-T [52,53,54] since it is a unique tumor-specific neo-antigen. However, it is not only present in less than 30% of GB patients [28,34,51], but also in a small number of tumor cells within individual tumors [34]. Several reports have indicated its value as a therapeutic target in various experimental designs [53,54,55,56]. However, some obstacles are yet to be overcome. Jiang et al. demonstrated that this problem may have arisen from intratumoral heterogeneity and that expression of EGFRvIII is detected only in a subpopulation of tumor cells and is lost over time [57]. Fry et al. also suggested that diminished antigen site density (CD19/CD22) is crucial and is even more important than total loss of its expression for achieving a therapeutic effect [58]. Our data suggest that these criteria, such as the specificity and effectiveness of EGFRvIII recognition and bounding, are fulfilled by the L8A4 antibody, which specifically binds with a characteristic EGFRvIII epitope. We compared the abovementioned antibody with another, A10, regarding binding to total EGFR protein. Although the specificity of the L8A4 antibody was confirmed, we cannot conclude that there is a complete lack of cross-reactivity with wild-type EGFR, which is abundantly expressed in normal cells.

Importantly, EGFRvIII flexibility in terms of disulfide bonds does not seem to hinder L8A4 EGFRvIII interaction.

Determining precisely how the density of such antigens as EGFRvIII influences cancer/tumor cells is difficult. The consequences of increasing antigen density as well as the percentage of GB cells showing EGFRvIII are uncertain. Theoretically, high antigen density can be responsible for adverse effects. Alternatively, antigen density as well as the percentage of EGFRvIII-positive cells or highly positive cells can be increased, and larger parts of or even the whole tumor can be eliminated. This is especially important in the case of EGFRvIII. EGFRvIII-positive cells frequently represent less than 30% of the cells in EGFRvIII-positive GB tumors. The situation seems to be even worse in colorectal, prostate, and breast cancer [28]. To this end, the search for methods that increase antigen density or even the percentage of EGFRvIII-positive cells within the existing number of cancer cells is important. Highly EGFRvIII-positive cells can represent epicenters of elimination in larger parts of GB.

## 4. Materials and Methods

### 4.1. Cell Culture

The AD293 cell line was purchased from Agilent Technologies (Santa Clara, CA, USA). Cells were cultured in Dulbecco’s modified Eagle’s medium DMEM (Biowest, Nauille, France). The DK-MG cell line was purchased from Leibniz-Institut DSMZ (Braunschweig, Germany) and cultured in RPMI 1640 (GIBCO, Life Technologies, Waltham, MA, USA). The MDA-MB-468 cell line was purchased from ATCC (USA) and cultured in DMEM/F12 (Biowest, Nauille, France). The NCI-H1975 cell line was purchased from ATCC (USA) and cultured in RPMI 1640 (GIBCO, Life Technologies, Waltham, MA, USA). The media used for culturing stable cell lines were supplemented with 10% FBS (Biowest, Nauille, France), 1% penicillin/streptomycin (Life Technologies, Waltham, MA, USA), and 0.2% gentamycin (Biowest, Nauille, France). Cells were maintained under standard cell culture conditions (5% CO_2_, 16% O_2_, 37 °C). Analyzed cell lines were in the early stages of passage and authenticated using NGS, MLPA, and Sanger sequencing methods.

The establishment of stable GB cell lines was performed as previously described [44]. Briefly, glioma cells were cultured in NSC-like conditions: a 1:1 mixture of Neurobasal medium (Life Technologies, Waltham, MA, USA) and DMEM/F12 (Biowest, Nauille, France), supplemented with N2, B27, antimitotic–antimycotic, Glutamax (all Life Technologies, Waltham, MA, USA), NEAA (Biowest, France), bFGF (40 ng/mL, Peprotech, London, United Kingdom), and EGF (5 ng/mL, Peprotech, London, UK). All samples were collected based on the protocol approved by the Bioethical Committee of the Medical University of Lodz (no. RNN/156/20/KE). Written informed consent was obtained from all patients, and their data were processed and stored according to the principles expressed in the Declaration of Helsinki.

### 4.2. Construction of Plasmids and Site-Directed Mutagenesis in EGFRvIII Transgene

The pLV1-puro-DEST vector was prepared as previously described [16,34,59]. Electrophoretic analysis and DNA sequencing were performed to verify the resulting recombinant vectors. The EGFRwt and EGFRvIII cDNA sequences were obtained, and expression plasmids were created [34,59]. The coding sequences were transferred to pLV1-puro-DEST under the CMV promoter via the LR reaction (Invitrogen, Waltham, MA, USA). Both sequences were confirmed with an Applied Biosystems 3130 Genetic Analyzer. The transfection was performed with Fugene HD (Promega, Madison, WI, USA) and puromycin (InvivoGen, San Diego, CA, USA), which were used to select cells that had successfully incorporated the plasmid.

Site-directed mutagenesis was performed on pENTR/EGFRvIII as previously described [59]. After the Gateway LR reaction and incorporation of the mutated EGFRvIII cassettes into pLV-puro-DEST vectors, all sequences were confirmed using the Applied Biosystems 3130 Genetic Analyzer. The primers used for the creation of cysteine–serine EGFRvIII mutants are listed in Appendix A.

### 4.3. Preparation of Genetic Content Delivery Vehicle and Establishment of Cell Lines

Stable AD293 cell lines expressing EGFRvIII mutations (AD293^C16S^, AD293^C20S^, AD293^C35S^, AD293^C38S^, and AD293^C42S^) as well as DK-MG^vIII-exovIII^ were obtained by transduction with lentiviruses created using the LENTI Smart Kit (InvivoGen, San Diego, CA, USA) as previously described [44]. The AD293^wt^ and AD293^vIII^ cell lines were established by transfecting the AD293^parental^ line with pLV-puro-EGFRwt and pIRESneo3-EGFRvIII using Fugene HD (Promega, Madison, WI, USA), selecting cells with puromycin or neomycin, respectively (both from InvivoGen, San Diego, CA, USA), and establishing a monoclonal population, as previously described [34]. DK-MG^extralow^, DK-MG^low^, and DK-MG^extrahigh^ were obtained by clonal selection of the parental DK-MG line through an assortment of cells with different EGFRvIII expression levels.

### 4.4. Reverse-Transcription Real-Time PCR

For the evaluation of EGFRwt and EGFRvIII expression levels in the analyzed cell models (primary and commercially available cell lines) as well as in the obtained DK-MG and AD293 sublines, nucleic acid isolation, reverse transcription, and real-time PCR were performed as previously described [28,44,59].

### 4.5. Compounds

The following concentrations of tyrosine kinase inhibitors were used: 0.5 µM afatinib, 10 µM erlotinib, and 10 µM lapatinib (unless other concentrations are defined), all purchased from Selleck Chemicals (USA). The dissolvent DMSO (Sigma-Aldrich, St. Louis, MI, USA) was used as a control. EGF (Invitrogen, Waltham, MA, USA) was used at a concentration of 20 ng/mL. Sodium orthovanadate (SOV; 1 mM; Calbiochem, Burlington, MA, USA) was used in Western blot analysis. L8A4 antibody (Kerafast, Boston, MA, USA) was added to the cell culture medium in the amount of 10 µg/mL to assess its interaction with EGFRvIII dimers.

### 4.6. Western Blotting

Denaturing blots were performed as previously described [34] with antibodies listed in Appendix A. To analyze receptor dimerization, confluent AD293 or DK-MG cells were harvested using a cell scraper, washed twice, and suspended in HBSS with ions. The indicated compounds were added to the cell suspension with incubation for one hour, or in the case of monolayer culture conditions; the medium was replaced by a fresh medium with the indicated compounds for one hour, washed twice with HBSS with ions, and harvested using a cell scraper. Thereafter, the cells were lysed in a cell lysis buffer supplemented with phosphatase and a Protease Inhibitor Cocktail (Sigma-Aldrich, St. Louis, MI, USA). Protein concentration was assessed using the Pierce BCA assay (Thermo Scientific, Waltham, MA, USA), non-reducing Laemmli buffer was added, and samples were boiled and then separated on 6%/12% gradient gel by SDS-PAGE electrophoresis. After transfer onto the PVDF membrane (Bio-Rad Laboratories, Hercules, CA, USA), the membrane was treated with 5% PhosphoBlocker (Cell Biolabs Inc., San Diego, CA, USA) and incubated with appropriate primary and HRP-conjugated secondary antibodies (Appendix A). Bands were visualized using Amersham ECL Prime Western Blotting Detection Reagent (GE Healthcare, Chicago, IL, USA) or the Opti-4CN Substrate Kit (Bio-Rad Laboratories, Hercules, CA, USA). Densitometry analyses were performed in ImageJ software.

### 4.7. Isolation of Protein Fractions

To estimate the EGFRvIII distribution in cells after treatment with particular TKIs, the isolation of protein fractions was performed according to the manufacturer’s protocol for the Subcellular Protein Fractionation Kit for Cultured Cells (Thermo Scientific, Waltham, MA, USA). Briefly, AD293^vIII^ cells were seeded onto 6-well plates in quantities of 3 × 10^5^ and left for 24 h to adhere. After that, the medium was changed to serum-free for another 24 h. Subsequently, DMSO and erlotinib (10 µM) were added to cells for 1 h in a fresh portion of serum-free medium. Then, cells were harvested, and particular protein fractions were isolated. For further analysis, Western blotting was conducted with 20 µg of protein fraction and an L8A4 antibody.

### 4.8. Immunofluorescence Analyses

Immunocytochemical staining was performed as previously described [44]. Briefly, cells were seeded onto 4-well plates in a complete medium. After 24 h, the medium was replaced with serum-free medium, followed by incubation for another 24 h. Afterward, the medium was replaced, and a fresh medium containing the examined compounds was added with incubation for 1 h. Then, cells were fixed and permeabilized, and nonspecific binding sites were blocked using 2% donkey serum (Sigma-Aldrich, St. Louis, MI, USA) in PBS with incubation for 1 h and gentle agitation. For double immunolabeling, fixed cells were subsequently incubated with the appropriate primary antibodies (1 h, RT) and further visualized by simultaneous incubation with a combination of species-specific fluorochrome-conjugated secondary antibodies (1 h, RT). Control samples were incubated with the secondary antibodies alone or otherwise processed identically. The slides were mounted with ProLong^®^ Gold Antifade Reagent with DAPI (Invitrogen, Waltham, MA, USA), coverslipped, and examined using an OPTA-TECH MN 800 fluorescence microscope. The fluorescence signal intensity was measured and compared between samples in ImageJ software. All antibodies utilized in immunocytochemical staining are indicated in Appendix A.

### 4.9. Computational Modeling

The sequence of EGFRvIII was obtained by computational modification of the EGFRwt sequence downloaded from the UniProt database. The modification started with the deletion of signaling peptides (positions 1–25), the deletion of 267 amino acids (positions 6–273 of the resulting sequence), followed by the insertion of a novel glycine residue at position 5.

Structural models of the EGFRvIII ectodomain (domains I–IV) with different rearrangements of disulfide bridges were created as comparative models based on chain A of the 3NJP template (dimer of the EGFRwt extracellular domain in its active conformation) using MODELLER software [38]. Disulfide bridges were explicitly and separately defined for each model as special MODELLER patches.

A sequence-level meta-analysis was performed using six predictive algorithms. The programs predict the state of cysteines (bonded or free-non-bonded); the prediction is supported by models that were created by machine learning methods applied to known crystal structures from the PDB database. The algorithms differ in the details of the algorithmic approach used to learn and make decisions. DiANNA [39] and SCRATCH [40], as well as DBCP [41] and CYSCON [42], were used to predict binding between cysteines, whereas DICON [42] and DISULFIND [43] only predict the cysteine state. DBCP additionally determines the likelihood of cysteine bonding with metals.

### 4.10. Statistical Analysis

The studies were performed at least in triplicate and presented as the average values ± standard deviation. The differences between samples were compared using Student’s two-tailed test and considered statistically significant at a *p*-value of less than 0. Differences between examined cell culture conditions were assessed by a two-way ANOVA. Results were considered statistically significant at *p* less than 0.05.

## 5. Conclusions

We demonstrated here that the application of TKIs such as afatinib or erlotinib (except lapatinib) increases the accumulation of EGFRvIII protein in cells; thereby, one can speculate that EGFRvIII bioavailability is increased for potential application in immunotherapeutics with improved outcomes. Hence, simultaneous implementation of EGFRvIII-specific CAR-T and TKIs may enhance the efficacy of immunotherapy and may constitute a viable and powerful approach with potential applications in CAR-T-based therapy. Moreover, existing different structures of EGFRvIII can still be recognized by immunotherapies, including CAR-T cells, based on L8A4. Only through more advanced analyses it can be tested whether this will result in a beneficial effect. One prediction is that high antigen density is responsible for adverse effects. Alternatively, not only is the antigen density increased, but the percentage of EGFRvIII-positive cells also increases, and CAR-T can be provoked to eliminate a larger proportion of the tumor. This seems to be more likely in the case of EGFRvIII. EGFRvIII-positive cells frequently represent less than 30% of the cells in EGFRvIII-positive GB tumors and do not seem to be tumor stem cells. The search for methods that increase the percentage of EGFRvIII-positive cells within those types of GB is important since methods for specifically eliminating those cells are being developed. The consequences of these changes should be more thoroughly characterized.

## Figures and Tables

**Figure 1 ijms-24-04350-f001:**
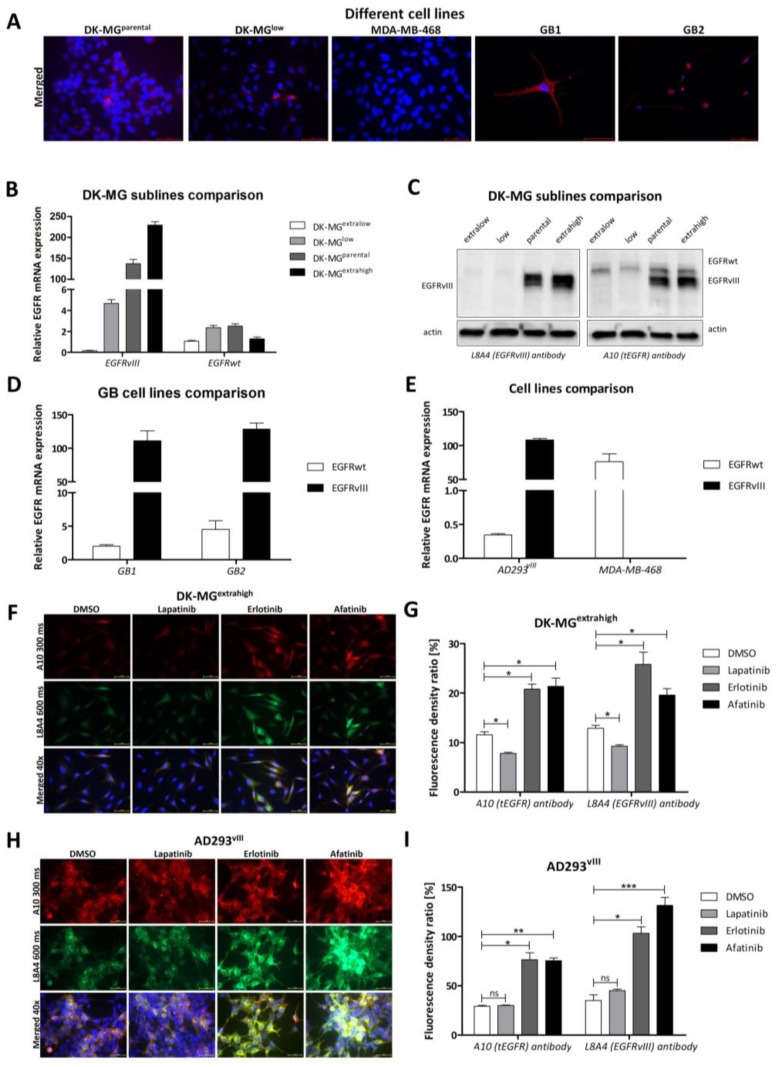
Characterization of research model according to different EGFRvIII expression profiles. (**A**) The cell lines DK-MG^parental^, DK-MG^low^, MDA-MB-468, and primary GB1 and GB2 culture were selected for testing L8A4 specificity in recognition and binding to the EGFRvIII epitope. For this purpose, cells were immunocytochemically stained with mouse L8A4 antibody (red) (**A**). There was no positive signal in the case of MDA-MB-468 cells, in which there is no EGFRvIII expression. About 5–10% and 50% of DK-MG^low^ and DK-MG^parental^ cell populations, respectively, interact with the L8A4 antibody, which corresponds with their profile of EGFRvIII expression. Specific binding of the antibody was also observed in primary GB cultures, with a previously confirmed expression of EGFRvIII. Taken together, these results demonstrate that the L8A4 antibody specifically recognizes EGFRvIII. Observations were performed under 40× magnification, except for GB2-20x, for the following exposition times: DAPI, 10 ms; L8A4, 600 ms. (**B**,**C**) Confirmation of EGFR (wt and vIII) expression levels in extralow, low, parental, and extrahigh DK-MG sublines performed at both mRNA (**B**) and protein (**C**) levels. Results for real-time PCR of (**D**) GB1 and GB2 and (**E**) MDA-MB-468 compared with AD293^vIII^. (**F**,**G**) The influence of particular TKIs (10 µM erlotinib, 0.5 µM afatinib, and 10 µM lapatinib) on total EGFR amount was investigated by immunocytochemical staining of DK-MG^extrahigh^ cell line. In both analyzed cases, erlotinib and afatinib increased, whereas lapatinib decreased the accumulation of EGFR protein. The fluorescence density ratio was measured in DK-MG^extrahigh^ using ImageJ software (**E**). (**H**,**I**) The influence of particular TKIs (10 µM erlotinib, 0.5 µM afatinib, and 10 µM lapatinib) on total EGFR amount was also investigated by immunocytochemical staining of AD293^vIII^ cell line. Signals for the A10 (total EGFR) and the L8A4 (EGFRvIII-specific) antibodies were also compared. ICC on AD293^vIII^ shows that erlotinib and afatinib increased whereas lapatinib decreased the accumulation of EGFRvIII protein (**H**). The fluorescence density ratio was measured using ImageJ software (**I**). Statistical significance was calculated using a two-tailed paired *t*-test from three independent experiments, ns > 0.05, * *p* ≤ 0.05, ** *p* ≤ 0.01, and *** *p* ≤ 0.001. Observations were performed under 40× magnification for the following exposition times: DAPI, 10 ms; L8A4, 600 ms; and A10, 300 ms.

**Figure 2 ijms-24-04350-f002:**
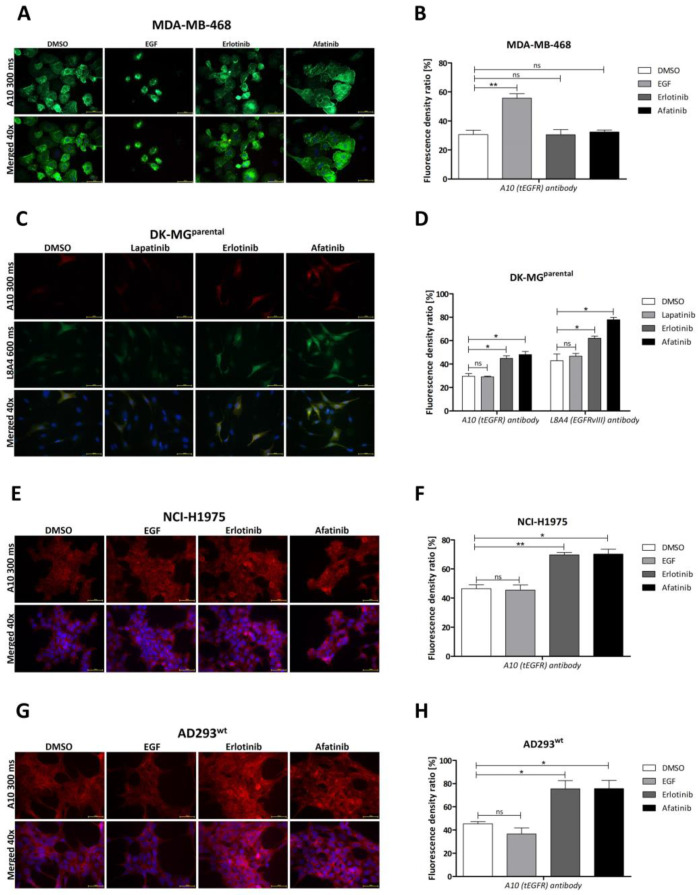
Characteristics of cell lines with different EGFR expression profiles. (**A,B**) An influence of particular TKIs—10 µM erlotinib and 0.5 µM afatinib as well as EGF 20 ng/mL on total EGFR amount was investigated by immunocytochemical staining on MDA-MB-468 cell line. In this case erlotinib and afatinib no significant effect on the EGFRwt accumulation was observed, whereas EGF increased it (**A**). The fluorescence density ratio was measured using ImageJ software (**B**). (**C**,**D**) The influence of particular TKIs (10 µM erlotinib, 0.5 µM afatinib, and 10 µM lapatinib) on total EGFR amount was also investigated by immunocytochemical staining of DK-MG^parental^ cell line (**C**). Similarly to DK-MG^extrahigh^, the EGFR signal in DK-MG^parental^ was increased by erlotinib and afatinib, whereas lapatinib decreased the accumulation of EGFR protein. The fluorescence density ratio was measured using ImageJ software (**D**). (**E**,**F**) The influence of particular TKIs (10 µM erlotinib, 0.5 µM afatinib, as well as EGF 20 ng/mL) on total EGFR amount was also investigated by immunocytochemical staining of the H1975 cell line. ICC on H1975 shows that erlotinib and afatinib increased, whereas lapatinib decreased the accumulation of EGFR protein (**G**). The fluorescence density ratio was measured using ImageJ software (**H**). (**G**,**H**) Influence of particular TKIs (10 µM erlotinib, 0.5 µM afatinib, as well as EGF 20 ng/mL) on total EGFR amount was also investigated by immunocytochemical staining of the AD293^wt^ cell line. ICC on AD293wt shows that erlotinib and afatinib increased whereas lapatinib decreased the accumulation of EGFRwt protein (**G**). The fluorescence density ratio was measured using ImageJ software (**H**). Statistical significance was calculated using a two-tailed paired *t*-test from three independent experiments: ns > 0.05, * *p* ≤ 0.05 and ** *p* ≤ 0.01. Observations were performed under 40× magnification for the following exposition times: DAPI, 10 ms; L8A4, 600 ms; A10, 300 ms; and sc03, 300 ms.

**Figure 3 ijms-24-04350-f003:**
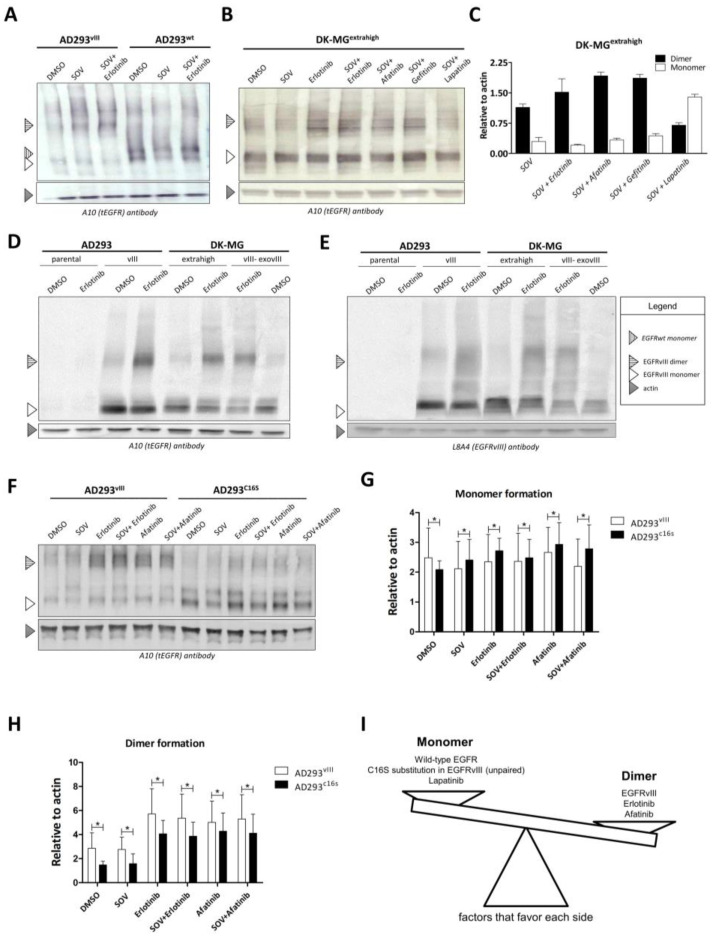
Mechanism of EGFR dimerization. (**A**) Semi-native Western blot demonstrating that EGFRvIII dimerization occurs via a homodimerization process that is not dependent on interaction between EGFRvIII and EGFRwt monomers. The addition of 10 µM erlotinib for 1 h further increased the dimerization of EGFRvIII. (**B**,**C**) Semi-native Western blots were performed using tEGFR antibody with prior treatment of SOV and TKIs: erlotinib (10 µM), afatinib (0.5 µM), gefitinib (10 µM), and lapatinib (10 µM) (**B**). Afatinib and gefitinib demonstrate similar activity to erlotinib, whereas lapatinib significantly decreases the stability of EGFRvIII homodimers. Using ImageJ, bands were quantified via densitometry and normalized to actin (**C**). (**D**,**E**) Semi-native Western blot with the parallel use of A10 (**D**) and L8A4 (**E**) antibodies in lines with different EGFRvIII expression profiles (endogenous DK-MG^extrahigh^, exogenous DK-MG^vIII-exovIII^, and AD293^vIII^) or with no expression of EGFRvIII-AD293 parental was performed to verify the potential differences in dimerization and antibody binding. The results indicate the same level of dimerization independent of the expression profile. Differences in EGFRvIII expression also do not affect the binding of antibodies. The addition of 10 µM erlotinib increased dimer formation. (**F**) Substitution of cysteine with serine in position 16 of EGFRvIII protein expressed in AD293 cell line significantly decreased the number of formed dimers and indicated that this cysteine is involved in disulfide bridge formation. In both cases, EGFRvIII and C16S, treatment of cells with TKIs such as 10 µM erlotinib or 0.5 µM afatinib for 1 h enhanced and stabilized dimerization. However, the addition of 1 µM SOV for 1 h alone or in combination with the abovementioned TKIs had no effect on this phenomenon, indicating that the dimerization process is not dependent on the phosphorylation status of EGFRvIII. (**G**,**H**) Statistical analysis of monomer (**G**) and dimer (**H**) formation in the case of AD293^vIII^ and AD293^c16s^ cells lines incubated by 1 h with 10 µM erlotinib, 0.5 µM afatinib, and 1 µM SOV (alone or in combination with the abovementioned TKIs) demonstrates that C16S substitutions impede dimer formation and favor the formation of monomers. (**I**) Schematic representation of factors that favor the formation of monomers (wild-type EGFR, C16S substitution, and lapatinib treatment) or dimers (EGFRvIII, erlotinib, and afatinib). Densitometric analysis (**C**,**H**,**I**) was obtained from three independent experiments, calculated by ImageJ, and data were analyzed using a two-tailed paired *t*-test, * *p* ≤ 0.05.

**Figure 4 ijms-24-04350-f004:**
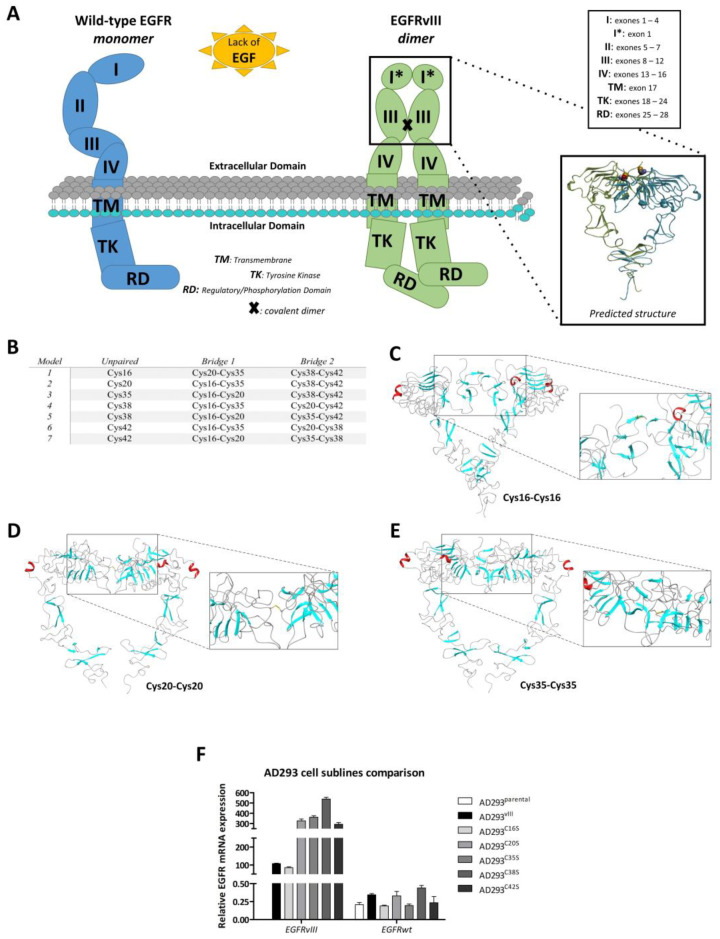
In silico analysis of dimer formation and the discovery of other cysteines were involved in this process in in vitro model development. (**A**) Monomer and dimer schematics. (**B**) All predicted disulfide bonds at the time of cysteine–serine substitution. (**C**–**E**) Predicted structures of interacting monomers. (**F**) According to the in silico models, the different cysteine–serine mutants, as well as EGFRvIII constructs, were prepared by site-directed mutagenesis of EGFRvIII cDNA and subsequent transformation into the AD293 cell line. Confirmation of expression levels of EGFRvIII and specific EGFRvIII cysteine–serine (C16S, C20S, C35S, C38S, and C42S) in established AD293 cell lines at the mRNA level.

**Figure 5 ijms-24-04350-f005:**
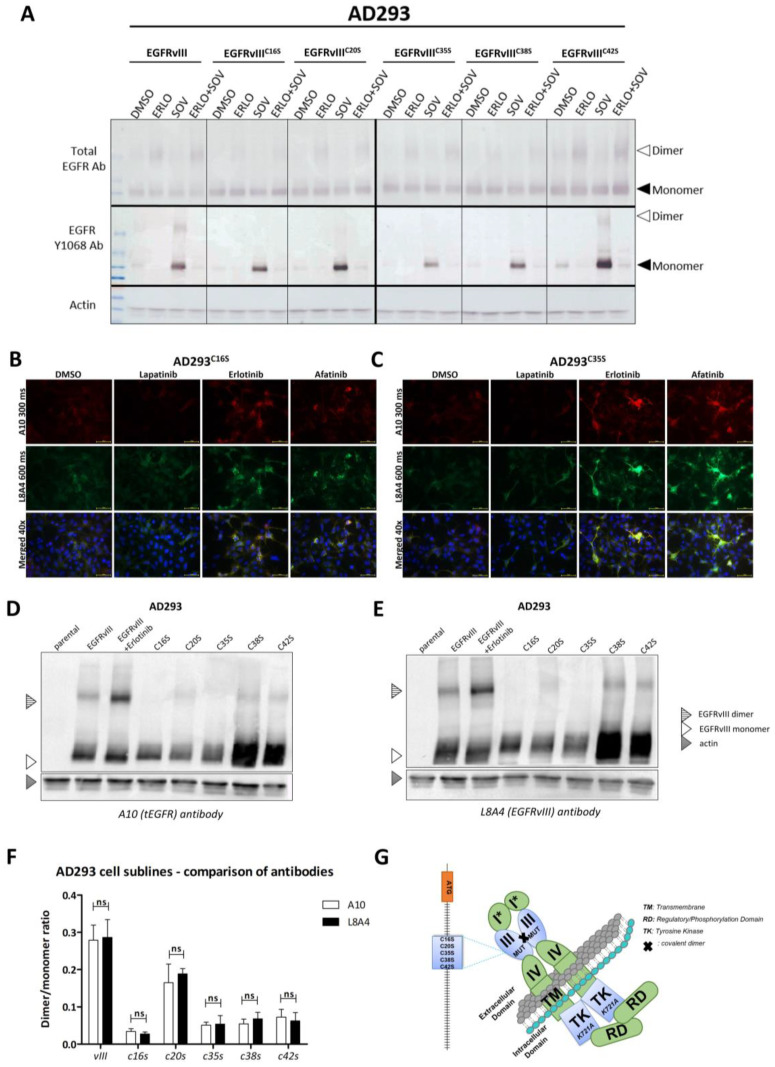
Validation of in silico models performed by in vitro testing of the different EGFRvIII cysteine–serine mutants. (**A**) Semi-native Western blot analysis of dimer formation ability in the case of AD293 cell line expressing EGFRvIII and EGFRvIII cysteine–serine mutants, C16S, C20S, C35S, C38S, and C42S in the presence of erlotinib, SOV, and a combination of both. (**B**,**C**) The influence of particular TKIs (10 µM erlotinib, 0.5 µM afatinib, and 10 µM lapatinib) on total EGFR amount was also investigated by immunocytochemical staining of AD293^C16S^ (**B**) and AD293^C35S^ (**C**) cell lines. In both cases, erlotinib and afatinib increased, whereas lapatinib decreased the accumulation of EGFRvIII protein. (**D**,**E**) Semi-native Western blot on AD293 parental, vIII, C16S, C20S, C35S, C38S and C42S cell lines was performed to elucidate whether different disulfide bridge arrangements may impede L8A4 antibody recognition of the EGFRvIII epitope (**D**). In addition to L8A4, the A10 antibody was tested (**E**). AD293^vIII^ cells were additionally treated with 10 µM erlotinib to verify the analysis. The results demonstrate that variables in interactions between EGFRvIII monomers do not impact on L8A4 mode of action. Moreover, there were no statistically significant differences in dimer creation when binding with either A10 or L8A4 antibodies (**F**). WB from (**D**,**E**) was calculated by ImageJ from three independent experiments, and data were analyzed using a two-tailed paired *t*-test, ns *p* > 0.05. (**G**) Schematic of introduced mutations. MUT: any of the following substitutions of cysteine by serine in positions 16 (C16S), 20 (C20S), 35 (C35S), 38 (C38S), or 42 (C42S). TM—transmembrane; TK—tyrosine kinase; RD—regulatory/phosphorylation domain.

## Data Availability

All data generated or analyzed during this study are included in this published article and its Appendix A. P.R. or C.T. should be contacted if someone wants to request the data.

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
