# Peer review of "Increased EGFRvIII Epitope Accessibility after Tyrosine Kinase Inhibitor Treatment of Glioblastoma Cells Creates More Opportunities for Immunotherapy"

_ijms, 2023, doi:10.3390/ijms24054350_

Round 1
Reviewer 1 Report
The authors look at the impact of both EGFRi and cysteine mutations on EGFRvIII dimer formation. I am not qualified to comment quality and significance of the computer modelling but the major concern with the rest of the manuscript is originality.
The effect of EGFRi (e,g erlotinib) on EGFRvIII dimerization and cell surface retention was described many years ago and has been verified by many groups. Likewise the effect of the C16S mutation has been documented by multiple groups over the past 15 years, The significance of the other cysteine mutations described is not clear to me. The C16 occurs due to the deletion event found in EGFRvIII and therefore is logical to study. The authors do not present any data showing that the cysteines in this manuscript have any biological relevance.
Specific concerns:
1) Reference 16 in the manuscript, not reference 13 was the first to describe the C16S. The authors make this mistake multiple times.
2) The pEGFR blot (Fig 4A) is unconvincing and questions the intepretation of the data. Previous work has shown that the dimer is the predominate phosphorylated species, but this suggests it is the monomer. Indeed, there is no phosphorylated dimer in the EGFRvIII control, this is simply wrong.
3) The constant reference to CAR T-cells is not relevant to the manuscript exceptt maybe a single line in the discussion.
Author Response
Please see the attachment.
We are extremely grateful for the constructive reviews.

Reviewer 2 Report
The major finding of this work is that EGFR inhibitors promote an apparent formation of truncated EGFRvIII dimers, whose activities are relevant to glioblastomas and other cancers. This is supported by experiments showing a shift in apparent molecular weight in WB studies and an increase for EGFRvIII immunoreactivity in the ICH experiments. The Authors perform additional in silico studies in an attempt to find a mechanistic explanation. Cys16 is reported as the most probable free cysteine for EGFRvIII dimerization (and activation). Indeed, the C16S but also C20S, C35S and C38S mutants inhibit dimerization in the presence of RTK inhibitors.
Overall, this study is extremely interesting (but not without caveats).
Here are some suggestions for the Manuscript revisions:
[1] Non-standard abbreviation ‘TKI’ should be spelled out in the title.
[2] For the General reader, CAR-T abbreviation should be spelled out in the abstract and explained in the text.
[3] The inference for improved CAR-T efficacy should be removed from the title because no direct experimental support for this idea is available in the current MS or existing publications.
[4] I think that the number of 40-50% of GBM cases, which the Authors associate with the EGFR gene amplification, is a combination or EGFR overexpression and aberrant activation. Please double-check.
[5] I did not check the relevance of all references, but it seems that the major ideas are occasionally supported by ‘fringe’ citations. The Authors need to check their references and opt in for quoting major primary publications. One example is the use of methodological publication of Foty et al ([23]) for making a number of bold conceptual statements such as “a huge potential in EGFRvIII targeting in vivo GB studies” (lns 85-86). This is incorrect and unacceptable.
[6] The Authors excessively refer to supplemental figure in discussing major findings in the text. If supplemental information is critical for conclusions, please move the relevant figures to the MS body.
[7] I think that promissory statements about the therapeutic potential of RTK inhibitors for GBM treatment should be removed because no experimental evidence in support of such conclusion is provided in the text.
[8] It seem that the Authors interpret increased IHC signal as an increase in the surface EGFRvIII levels. However, their IHC studies have been performed in permeabilized cells. The text and conclusions should be double checked against this comment. This is a major caveat, including for the inferences about changes in efficacy of CAR-T.
Author Response

(The authors gave the same response as above.)

Reviewer 3 Report
The authors describe herein a favorable space for CAR-T-based therapies following an increase in EGFRvIII epitope accessibility due to TKI treatment of glioblastoma cells. The paper is consistent, but some things need to be made clearer before recommending the manuscript for publication.
Major:
1. The authors should be more consistent regarding their justification for the selected cell line or explain more clearly on why they choose to use specific cell lines only in some of their experiments. For example, Figure 1A shows parental DK-MG and only DK-MG low subpopulation but not the other two, very low and very high, which are entered only at points B and C.
2. In Figure 1C the authors should mention either on the figure or in the figure legend the differences between the left and right panels, most likely membrane probing with L8A4 and A10 antibodies. It would be good to add the densitometric values ​​of the bands for a better visualization and comparison with the mRNA results from B.
3. Please provide for the bands detected by Western Blot shown in Figure 2, the quantitative measurements as in point C and specify the number of replicates for WB.
Please provide for all Western Blot the corresponding densitometry analysis and indicate the number of replicates and the statistical test used.
Minor:
1. P2, L69 please modify „verified” with „verify”.
2. P5, L170: Probably should be “… which requires careful analysis…“ instead of “… what requires higher care while…”.
3. Please avoid unnecessary repetition: “The algorithms differ in the details of the algorithm approach…”.
Author Response

(The authors gave the same response as above.)

Reviewer 4 Report
The manuscript by Treda et al. provides evidence supporting beneficiary use of EGFRvIII and TKI. They also showed that the L8A4 antibody recognizes both WT and mutated forms of EGFR in mono and dimer formats. Although the manuscript does not use many techniques to support the author's claim and hypothesis, the results are sufficient for publication. However, I have just a few comments to improve the quality of the manuscript further: 1) Figure 2 D and E use the same actin for both gels.2) To confirm their dimerization model, authors could have a construct of all five cysteine mutations and show the dimerization interruption.
3) Although authors show that the L8A4 antibody performed well, this doesn't mean this will perform well in a CAR structure. Every scFv will behave differently, which highly depends on the CAR design, such as the choice/length of the hinge region, TM region, amount of CAR expression, and the co-stimulatory domain. Years of working in this field showed me that not all Ab would work in the CAR context, as tonic signaling cannot be predicted unless CAR T cells are manufactured. I suggest authors focus more on broader terms, such as immunotherapy, in their title as well as throughout the manuscript. It is Ok if they recommend the potential use of this antibody in the CAR format. The last sentence of the discussion needs to be corrected. High antigen density is responsible for higher CAR T cell expansion which can lead to AE.Author Response
Please see the attachment.
We are extremely grateful for the constructive reviews.

Round 2
Reviewer 1 Report
The authors have adequately addressed my concerns.